# Increased Levels of BAMBI Inhibit Canonical TGF-β Signaling in Chronic Wound Tissues

**DOI:** 10.3390/cells12162095

**Published:** 2023-08-18

**Authors:** Sabrina Ehnert, Helen Rinderknecht, Chao Liu, Melanie Voss, Franziska M. Konrad, Wiebke Eisler, Dorothea Alexander, Kristian-Christos Ngamsri, Tina Histing, Mika F. Rollmann, Andreas K. Nussler

**Affiliations:** 1Siegfried Weller Research Institute, BG Unfallklinik Tübingen, Department of Trauma and Reconstructive Surgery, University of Tübingen, Schnarrenbergstr. 95, 72076 Tübingen, Germanyweisler@bgu-tuebingen.de (W.E.); mrollmann@bgu.tuebingen.de (M.F.R.); andreas.nuessler@gmail.com (A.K.N.); 2Department of Anesthesiology and Intensive Care Medicine, University Hospital of Tübingen, Hoppe-Seyler-Straße 3, 72076 Tübingen, Germany; franziska.konrad@med.uni-tuebingen.de (F.M.K.); kristian.ngamsri@med.uni-tuebingen.de (K.-C.N.); 3Department of Oral and Maxillofacial Surgery, University Hospital Tübingen, Osianderstr 2-8, 72076 Tübingen, Germany; dorothea.alexander@med.uni-tuebingen.de

**Keywords:** wound healing, chronic wounds, TGF-β, BAMBI, angiogenesis

## Abstract

Chronic wounds affect more than 2% of the population worldwide, with a significant burden on affected individuals, healthcare systems, and societies. A key regulator of the entire wound healing cascade is transforming growth factor beta (TGF-β), which regulates not only inflammation and extracellular matrix formation but also revascularization. This present work aimed at characterizing wound tissues obtained from acute and chronic wounds regarding angiogenesis, inflammation, as well as ECM formation and degradation, to identify common disturbances in the healing process. Serum and wound tissues from 38 patients (N = 20 acute and N = 18 chronic wounds) were analyzed. The patients’ sera suggested a shift from VEGF/VEGFR to ANGPT/TIE2 signaling in the chronic wounds. However, this shift was not confirmed in the wound tissues. Instead, the chronic wound tissues showed increased levels of MMP9, a known activator of TGF-β. However, regulation of TGF-β target genes, such as *CTGF*, *COL1A1*, or *IL-6,* was absent in the chronic wounds. In wound tissues, all three TGF-β isoforms were expressed with increased levels of TGF-β1 and TGF-β3 and a reporter assay confirmed that the expressed TGF-β was activated. However, Western blots and immunostaining showed decreased canonical TGF-β signaling in the respective chronic wound tissues, suggesting the presence of a TGF-β inhibitor. As a potential regulatory mechanism, the TGF-β proteome profiler array suggested elevated levels of the TGF-β pseudo-receptor BAMBI. Also, tissue expression of *BAMBI* was significantly increased not only in chronic wounds (10.6-fold) but also in acute wounds that had become chronic (9.5-fold). In summary, our data indicate a possible regulatory role of BAMBI in the development of chronic wounds. The available few in vivo studies support our findings by postulating a therapeutic potential of BAMBI for controlling scar formation.

## 1. Introduction

Chronic wounds represent a significant burden not only on the affected individuals but also on the healthcare system and society. A recent meta-analysis postulated a 2.2% prevalence of chronic wounds worldwide—which is still a conservative estimate considering that only a few reports from developing countries could be included [1]. Worldwide, the mean annual costs for treating a leg, pressure, or diabetic foot ulcer lay between US 11,000 to US 44,200 [2], resulting in a substantial economic burden for the healthcare systems.

Wound healing is a dynamic and complex process. tightly orchestrated by different growth factors, cytokines, and chemokines. These factors control the initial inflammatory processes at the wound site, as well as the following resolution of inflammation associated with new vessel and matrix formation. Risk factors for the development of chronic, non-healing wounds are manifold, and include the patient’s age, physical constitution (e.g., diabetes mellitus, vascular deficits, or hypertension), and lifestyle (e.g., smoking) [3].

Many of the risk factors are associated with altered inflammation and dysregulation of transforming growth factor beta (TGF-β). TGF-β, the eponym of the TGF-β superfamily, is a dimeric, secreted polypeptide with three prototypic isoforms—TGF-β1, TGF-β2, and TGF-β3—that account for specific roles in wound healing. In normal wound healing, TGF-β (mainly TGF-β1) is released from platelet granules during the blood clotting cascade [4]. This rapid local increase in TGF-β1 is thought to guide immune cells, fibroblasts, and keratinocytes to the wound site [5]. There, TGF-β1 and TGF-β2 act as potent regulators of inflammatory responses, and their peak levels are usually associated with a resolution of inflammation [5]. However, local increases in TGF-β3 may have antagonistic effects on the immune cells [6]. As growth factors, all three TGF-β isoforms favor re-epithelialization of the wound tissue. Although the underlying mechanisms are not fully understood, all three TGF-β isoforms are reported to promote angiogenesis in the proliferative phase of wound healing [6]. Local increases in TGF-β are associated with increases in vascular endothelial growth factor (VEGF) levels, either by recruitment of hematopoietic effector cells, induction of VEGF expression, or a combination of both [7]. Further, all three TGF-β isoforms may induce epithelial-to-mesenchymal transition (EMT), which has been widely linked to the sprouting phase of angiogenesis [8]. While processes of the proliferative phase of wound healing are mainly promoted by all three TGF-β isoforms, the situation becomes different during the remodeling phase of wound healing. There, fibroblast-to-myofibroblast transition is controlled by all three TGF-β isoforms in a concentration-dependent manner. High levels of TGF-β1 and TGF-β2 are reported to favor the deposition of extracellular matrix (ECM) in the wound, which includes the expression of connective tissue growth factor (CTGF), collagen, or fibronectin [9]. Especially, TGF-β1 may also inhibit ECM remodeling by matrix metalloproteinases (MMPs) [10]. Thus, excessive or chronically elevated TGF-β1 may promote scar/keloid formation and fibrosis in adults. In contrast, low levels of TGF-β1 and TGF-β2 may have antagonizing effects. Further, high levels of TGF-β3 have been reported to reduce scarring in adults and even promote scarless healing in the fetus [11]. These examples show that TGF-β is a key regulator in the different phases of wound healing, affecting different cell types.

TGF-β ligands are secreted as inactive precursors bound to latency-associated peptides, which need to be activated by lytic enzymes, e.g., elastase or MMPs, to bind to their transmembrane receptors. TGF-β receptors, which are usually divided into three classes (TGFβRI, TGFβRII, and TGFβRIII), are the only cell surface serine/threonine kinase receptors known in humans. Active TGF-β ligands first bind to a dimer of the constitutively active TGFβRII. Upon additional binding to a dimer of TGFβRI (in most cell types ALK5 (activin-like kinase 5)), the proximity allows the TGFβRII to phosphorylate the TGFβRI [12]. Once activated, the tetrameric receptor complex initiates an intracellular cascade, which for the canonical TGF-β signaling includes the phosphorylation and nuclear translocation of Smads (typically Smad2/3).

As TGF-β has diverging roles in many different cells and tissues, a plethora of control mechanisms and inhibitors exist. In addition to the controlled expression of the receptors and transcription factors of the signaling pathway itself, the control mechanisms include the expression of required co-receptors (TGFβRIII: e.g., endoglin or betaglycan), co-factors (e.g., SARA (Smad anchor for receptor activation), SKI (Sloan–Kettering Institute), or SnoN (Ski novel)), but also the expression of the negative feedback inhibitor Smad7. Further, TGF-β signaling might be depleted by proteasomal degradation of Smads with the help of Smad ubiquitin regulatory factors (Smurf1 and 2). Other inhibitors expressed by the cells themselves include inhibin (interacts with betaglycan), FKPB12 (FK506-binding protein, interacts with ALK5), or the TGF-β pseudo-receptor BAMBI (BMP and activin membrane-bound inhibitor), also known as non-metastatic gene A (NMA) [13]. BAMBI has structural similarities to TGFβRI, which favors dimerization with the latter. These TGFβRI/BAMBI heterodimers compete with the TGFβRI dimers for receptor complex formation with TGF-β ligands and TGFβRII dimers. In contrast to TGFβRI, BAMBI lacks the intracellular kinase domain, which blocks the expected phosphorylation cascade at the receptor level. When first characterizing the function of BAMBI, it was demonstrated that the pseudo receptor effectively blocks both Smad2/3- and Smad1/5/8-dependent signaling [14]. Investigating protein interactions, this study further postulated no direct interaction of BAMBI with BMPs or activins, thus challenging the inhibitory effects of circulating BAMBI. Observations in BAMBI-/- mice suggest that BAMBI might inhibit not only the Smad-dependent canonical TGF-β signaling but also the Smad-independent non-canonical TGF-β signaling [15]. However, this requires further investigation.

These examples clearly demonstrate that TGF-β is an important regulator in wound healing and the angiogenic processes required for it. However, there are many regulatory pitfalls that can affect wound healing at different stages. Thus, the present work aimed to investigate the angiogenic potential and the associated TGF-β signaling in tissues of acute and chronic wounds more closely.

## 2. Materials and Methods

### 2.1. Ethics

Blood and wound tissues were obtained after receiving written informed consent from the donors, following the Declaration of Helsinki (1964) and its most recent amendments. Blood to produce in vitro scabs was obtained from healthy volunteers as approved by the local ethics committee (844/2020BO2). Serum and wound tissues were obtained from acute wounds and chronic wounds (primary and revision surgeries) surgeries as approved by the local ethics committee (666/2018BO2).

### 2.2. In Vitro Scabs

In vitro scabs were produced as previously described [16]. Briefly, mycoplasma-free HaCaT cells were expanded in culture medium (Dulbecco’s Modified Eagle’s Medium (DMEM)—high glucose with 5% FCS) at 37 °C, 5% CO_2_, and humidified atmosphere. For each in vitro scab 60 µL of a 0.5 × 10^6^ cells/mL HaCaT cell solution (DMEM—high glucose, 7.5 mM CaCl_2_, 5% FCS) was mixed with 60 µL fresh EDTA venous blood in sterile non-adherent 96-well plates with F-shaped bottom. After incubation (37 °C, 5% CO_2_, and humidified atmosphere) for 1 h, the coagulated in vitro scabs were transferred to fresh 96-well plates for stimulation with the three recombinant human (rh) TGF-β isoforms. The medium was replaced every 48 h.

### 2.3. Tube Formation Assay

Human umbilical vein endothelial cells (HUVECs) were expanded (37 °C, 5% CO_2_, and humidified atmosphere) on 0.1% Gelatin-coated flasks in Endothelial Cell Growth Basal Medium 2 (EBM-2) with 2% FCS, 1% Antibiotic/Antimycotic, 0.5 ng/mL h VEGFA165, 10 ng/mL h FGF-b, 5 ng/mL h EGF), 20 ng/mL h IGF-R3, 22.5 µg/mL heparin, 0.2 µg/mL hydrocortisone, 1 µg/mL L-ascorbic acid. 6.5 × 10^4^ HUVECs were seeded in 48-well plates and coated with a thin layer (6 µL) of Geltrex^TM^ (Thermo Scientific, Waltham, MA, USA) [17]. The conditioned media from the in vitro scabs were diluted 1:1 in plain EBM-2 and added to the cells. After 18 h of culture (37 °C, 5% CO_2_, and humidified atmosphere), cells were stained with 20 µM Calcein-AM for 10 min. Microscopic images were captured with the EVOS FL imaging system (Thermo Scientific, Waltham, MA, USA) and analyzed using the ImageJ angiogenesis analyzer [18].

### 2.4. Cytokine Arrays

Cytokine profiles from patients’ sera and wound tissues (N = 12 for each pool) were determined using the RayBio^®^ Human Angiogenesis Array C2 and the RayBio^®^ Human TGF-β Array 2 (BioCat, Heidelberg, Germany). The arrays were performed according to the manufacturer’s instructions. Chemiluminescent signals were detected with a CCD camera (INTAS, Göttingen, Germany) and quantified with ImageJ 1.47v (NIH, Bethesda, MD, USA). After background correction (blank values), all spots of one membrane were normalized to the respective positive controls. Targets with signal intensities below the standard error of the mean of the 6 positive controls (detection limit) were excluded from further analyses with JMP 16 (SAS Institute GmbH, Heidelberg, Germany).

### 2.5. Gene Expression Analyses

Total RNA was isolated by phenol–chloroform extraction and quantified using a spectrophotometer (Omega plate reader, BMG Labtech GmbH, Ortenberg, Germany). A maximum of 2.5 µg total RNA was used to synthesize cDNA with the First Strand cDNA Synthesis Kit (Thermo Fisher Scientific, Sindelfingen, Germany). Primer sequences and optimized RT-PCR conditions are summarized in Table 1. Relative mRNA expression levels were calculated with the geometric mean of *18s* and *RPL13a* as the most stable housekeeping genes. For qRT-PCR, the reaction efficiencies were confirmed to be between 85% and 115%. The specificity of the PCR reactions was checked by melting curve analyses and agarose gel-electrophoresis.

### 2.6. Immunostainings of Phosphorylated Smad3

For immunostainings, formalin-fixed wound tissues were embedded in paraffin for slicing with the microtome. The 3 µm thick slices were transferred to silanated object slides. Before staining, slides were heated (60 °C, 30 min), then de-paraffinized with Roticlear^®^ (3 × 7 min), and rehydrated in an ethanol gradient (100%, 96%, 70%—each 5 min). After washing with PBS antigen retrieval was performed in a 10 mM-sodium citrate solution (pH = 6.0) by heating in a microwave (750 W, 7 min). After cooling for 20 min at room temperature, unspecific binding sites were blocked (5% BSA and 1% normal donkey serum in PBS-T) for 1 h at room temperature.

#### 2.6.1. Immunofluorescence Staining with Confocal-Microscopy

After washing with PBS, the phospho-Smad3 (1D9) antibody (Santa Cruz Biotechnology, Heidelberg, Germany, #sc-517575 diluted 1:50 in 1% BSA and 1% normal donkey serum in PBS-T) was applied to the tissues for incubation overnight at +4 °C in a humidified atmosphere. Incubation with normal IgG (Santa Cruz Biotechnology #sc-2025) and pure diluent were used as background controls. After washing with PBS (4 × 5 min), a secondary antibody (AF488 donkey anti-mouse antibody, Invitrogen, Waltham, MA, USA, #A21202 diluted 1:500 in 1% BSA and 1% normal donkey serum in PBS-T) was applied and incubated for 40 min at room temperature. After washing with PBS (4 × 5 min), tissues were embedded into Roti^®^Mount FlourCare with DAPI (Carl Roth, Karlsruhe, Germany, #HP20.1). Fluorescent image recording was done with a confocal laser microscope LSM510 from Zeiss (Oberkochen, Germany) with the microscope and imaging software ZEN 2011 (black edition). Nuclear phospho-Smad3 was quantified by co-localization analysis in ImageJ 1.47v.

#### 2.6.2. Immunohistological Staining

A Dual Endogenous Enzyme Block (Agilent DAKO, Santa Clara, CA, USA, #S2003) was used to inhibit endogenous peroxidase activity. After washing with PBS, the phospho-Smad3C (Ser 423/425) antibody (IBL, Hamburg, GER, #28,031 diluted 1:50 in 1% BSA and 1% normal donkey serum in PBS-T) was applied to the tissues for incubation overnight at +4 °C in a humidified atmosphere. Each tissue section was also incubated with pure diluent as background controls. The next day, tissue sections were washed with PBS (4 × 5 min) and secondary antibody HRP-conjugated mouse-anti-rabbit IgG (Santa Cruz Biotechnology #sc-2357 diluted 1:200 in 1% BSA and 1% normal donkey serum in PBS-T) was applied for 60 min at room temperature. After washing with PBS (4 × 5 min), the ABTS substrate solution was applied, and tissue sections were incubated at room temperature for color development. After 10 min, tissue slices were washed with PBS (4 × 5 min) and counterstained with hematoxylin. After final dehydration and mounting, microscopic images were captured and analyzed with ImageJ.

### 2.7. Luciferase Reporter Assay

HaCaT cells were infected with either the Smad2/3 reporter adenovirus (Ad5-CAGA-Luc, kindly provided by Prof. Peter ten Dijke) or the Smad1/5/8 reporter adenovirus (Ad5-BRE-Luc, kindly provided by Dr. O Korchynskyi and Prof. Peter ten Dijke) in serum-free DMEM medium. The next day, the serum-free medium was refreshed, and cells were stimulated with 20% patients’ sera (10 ng/mL rhTGF-β1 or rhBMP7 as positive controls) for 48 h. To identify the inhibitory effects of BAMBI on both Smad-dependent pathways, the culture was additionally supplemented with rhBAMBI (#TP761788, OriGene Technologies, Rockville, MD, USA). Active TGF-β in the patients’ sera causes phosphorylation of Smad2/3, while active BMPs in the patients’ sera cause phosphorylation of Smad1/5/8. The phosphorylates Smads induce the expression of luciferase in the cells by binding to the CAGA/BRE reporter sequence. The luciferase activity in cell lysates was measured according to the manufacturer’s instructions, using the SteadyGlo Luciferase Assay System (Promega, Madison, WI, USA).

### 2.8. Western Blot

Wound tissues, immersed in RIPA buffer with protease and phosphatase inhibitors, were snap frozen at −80 °C until further use. After homogenization on ice, tissue debris was removed by centrifugation (14,000× *g*; 10 min; +4 °C). Then, 35 µg of each protein lysate, quantified by Lowry [19], was separated on a 12% SDS-PAGE gel and then transferred to a nitrocellulose membrane by wet-blot transfer. Protein transfer was confirmed by Ponceau S staining. After blocking the membranes (5% BSA in TBS-T) for 1 h, the incubation with primary antibodies was done overnight at +4 °C (phospho-Smad3 (1D9)/Santa Cruz Biotechnology #sc-517575 and GAPDH/Merk #G9545). After washing with TBS-T, membranes were incubated with the respective HRP-coupled secondary antibodies for 2 h at room temperature. Chemiluminescent signals were captured with an INTAS Chemocam (INTAS, Göttingen, Germany) and quantified with ImageJ.

### 2.9. Statistical Analysis

JMP 16 and GraphPad Prism 8.01 were used to perform the statistical analysis. Array data are visualized as heat maps and constellation plots. Semi-/Quantitative Data are presented as box plots with individual data points. The number of biological (N) and technical (n) replicates are given in the figure legends. Conditions were compared by the non-parametric Mann–Whitney U test or Kruskal–Wallis test, followed by Dunn’s multiple comparison test. A *p* < 0.05 was considered statistically significant.

## 3. Results

### 3.1. Patient Characteristics

Thirty-eight patients were included in the current study—their characteristics are summarized in Table 2. Wound tissue and serum were obtained from 18 patients with chronic wounds (persisting > 60 days at the time point of sampling) and 20 patients with acute wounds (persisting < 40 days at the time point of sampling)—of those, five showed no healing in the following 6 weeks. The average wound size at the time of sampling was comparable between the groups. Likewise, the distribution of sex and age of the donors, as well as the rate of confounding factors, e.g., diabetes mellitus and smoking were comparable between the groups. Chronic wounds were more often colonized with bacteria. However, this was not significant.

### 3.2. Patients with Chronic Wounds Show Altered ANGPT:TIE2 Ratios in the Serum and Increased Levels of Matrix Metalloproteinase 9 in the Wound Tissue

It is well-accepted that disturbed angiogenesis contributes to the development of chronic wounds. The RayBio^®^ Human Angiogenesis Array C2 was used to screen for angiogenic factors in both the sera and wound tissues of patients with acute (healing) and chronic wounds (Figure 1). In the patients’ sera, a large difference in the ratio between angiopoietins and their receptor TIE2 was observed. While serum from patients with acute wounds showed higher levels of ANGPT1 and ANGPT2, serum from patients with chronic wounds showed higher levels of TIE2. Furthermore, serum from patients with chronic wounds showed higher levels of MMP1. These differences were not detected in the respective wound tissues. In contrast, chronic wound tissues showed higher levels of MMP9 and pro-inflammatory interleukins IL-1β and IL-2. Meanwhile, acute wound tissues showed slightly elevated levels of endostatin and the urokinase-type plasminogen activator receptor (uPAR).

### 3.3. Increased Expression of FN1, VEGFR2, and Pro-Inflammatory Interleukins in Chronic Wound Tissue

The prior screening suggested an imbalance in matrix remodeling. Therefore, the expression of genes involved in matrix formation and degradation was analyzed with the individual tissue samples (Figure 2a). Increased expression of *MMP1* and *MMP9* in the chronic wound tissues was not significant. Expression of MMP tissue inhibitors *TIMP1* and *TIMP2* was also not significantly altered in the chronic wound tissues. Similarly, expression of *CTGF* and alpha-1 type I collagen (*COL1A1*) was not altered in chronic wound tissues. However, fibronectin (*FN1*) expression was significantly higher in chronic wound tissues compared to acute wound tissues (2.0-fold, *p* = 0.0232).

Further, the expression of angiogenic factors and their receptors was investigated. In line with the Angiogenesis Array, the expression of *VEGFa* and its receptor *VEGFR1* (*FLT*), as well as the expression of *ANGPT1* and its receptor *TIE2* (*TEK*) was comparable in acute and chronic wound tissues. In contrast to the Angiogenesis Array, the expression of *VEGFR2* (*KDR*: 4.9-fold, *p* = 0.0062) and *ANGPT2* (1.5-fold, *p* = 0.0630) was increased in chronic wound tissues when compared to acute wound tissues (Figure 2b).

Considering inflammatory processes, expression of interleukins *IL-1β* (4.6-fold, *p* = 0.0074), *IL-6* (2.5-fold, *p* = 0.0357), and *IL-8* (*CXCL8*: 2.3-fold, *p* = 0.0074) were increased in chronic wound tissues when compared to acute wound tissues (Figure 2c).

A string analysis was performed to show protein interactions (Figure 2d). This identified TGF-β as a key modulator of these proteins.

### 3.4. Increased Expression of TGF-β1 and TGF-β3 in Chronic Wound Tissues

TGF-β isoforms have been described as playing diverging roles in ECM formation, angiogenesis, and inflammation during wound healing. Gene expression of the three TGF-β isoforms was quantified in the patients’ wound tissues (Figure 3a). While expression of *TGF-β2* was comparable in both wound tissue types, chronic wound tissues expressed higher mRNA levels of *TGF-β1* (4.1-fold, *p* = 0.0112) and *TGF-β3* (1.7-fold, *p* = 0.0804).

Proangiogenic effects of the three TGF-β isoforms were confirmed by using an in vitro scab model (Figure 3b–d). Exposure to rhTGF-β1 (3.2-fold, *p* < 0.0001) and rhTGF-β2 (2.1-fold, *p* = 0.0070) significantly increased *VEGFa* expression in the in vitro scabs. This was not observed when in vitro scabs were exposed to rhTGF-β3 (1.4-fold) or the Alk5 inhibitor (SB431542/0.9-fold/Figure 3b). Factors secreted by the rhTGF-β exposed in vitro scabs favored tube formation in HUVECs. In vitro scabs exposed to rhTGF-β1 strongly induced tube formation (Figure 3c), as quantified by the largest mesh area, most junctions, and the highest branching interval. In contrast, inhibiting the TGF-β signaling in the in vitro scabs Alk5 inhibitor (SB431542) effectively inhibited tube formation in the following tube formation assay, as quantified by most isolated segments (Figure 3d).

### 3.5. Expressed TGF-β Is Active but Does Not Reach the Chronic Wound Tissues

The increased expression of *TGF-β1* and *TGF-β3* in chronic wound tissues should activate the TGF-β signaling in these tissues. However, TGF-β is normally expressed as an inactive precursor that requires activation. Therefore, by using an adenoviral-based reporter assay [20], the levels of active TGF-β were quantified in the respective patients’ sera (Figure 4a). In line with the increased gene expression of *TGF-β1* and *TGF-β3* in chronic wound tissues, serum levels of active TGF-β were significantly increased in patients with chronic wounds (1.6-fold, *p* = 0.0023).

Quantification of the phosphorylated Smad3 in the respective wound tissues by Western blotting (Figure 4b) revealed that phospho-Smad3 levels were significantly lower in chronic wound tissues compared to acute wound tissues (0.7-fold, *p* = 0.0060). In line with that, co-localization analysis of the confocal immunofluorescent images showed that less phospho-Smad3 entered the nucleus in chronic wound tissues (0.8-fold, *p* = 0.0312/Figure 4c,d). The reduced Smad3 phosphorylation and nuclear translocation in the presence of increased active TGF-β levels suggest the inhibition of TGF-β signaling in chronic wound tissues.

### 3.6. Reduced TGF-β Signaling in Chronic Wound Tissues

Further, an immunohistochemical staining of phosphorylated Smad3 was performed on the wound tissues, to investigate if the observed inhibition in TGF-β was restricted to specific wound areas (Figure 5). The staining showed an inhomogeneity of the examined wound tissues, with areas staining stronger or weaker for phospho-Smad3. However, quantification of the stained areas revealed significantly less phospho-Smad3 (0.5-fold, *p* = 0.0122) over all the chronic wound tissues.

### 3.7. Expression of the TGF-β Pseudo-Receptor BAMBI Is Increased in Chronic Wound Tissues

The RayBio^®^ Human TGF-β Array 2 was performed to screen for possible modulators of TGF-β signaling. Both the sera and wound tissues of patients with acute (healing) and chronic wounds were analyzed (Figure 6). In both the patients’ sera and wound tissues, most of the detected factors were comparable between both groups.

TGF-β1 and TGF-β2 showed the same trend as at gene expression level. BMP4, GDF11 (BMP11), and follistatin were slightly decreased in both sera and wound tissues of patients with chronic wounds. Noggin showed a similar trend but only in the patients’ sera. BMP2 and GDF15 (MIC-1) tended to increase only in the sera of patients with chronic wounds. Noteworthy, GDF15 (MIC-1) was barely detectable in any of the wound tissues. The TGF-β pseudo-receptor BAMBI (BMP and activin membrane-bound inhibitor) demonstrated a strong increase in both serum and wound tissue of patients with chronic wounds. The related constellation blot revealed that BAMBI levels were not associated with any of the other factors detected (Figure 6b).

As BAMBI can effectively inhibit TGF-β, its expression was confirmed in the patients’ wound tissues (Figure 6c). For this analysis, tissues from acute wounds with a following delay in the healing process were also included. As suggested by the TGF-β array, *BAMBI* expression was significantly increased in chronic wound tissues (10.6-fold, *p* = 0.0062) when compared to acute (healing) wound tissues. Furthermore, *BAMBI* expression was significantly increased in acute wounds with a later delay in the healing process (9.5-fold, *p* = 0.0149).

### 3.8. Both Smad-Dependent Signaling Pathways Are Inhibited in the Presence of BAMBI

The RayBio^®^ Human TGF-β Array 2 suggested that BAMBI levels were elevated not only in the chronic wound tissues but also in the sera of patients with chronic wounds. However, the TGF-β reporter assay showed stronger activation of the signaling cascade by these sera when compared to sera of patients with chronic wounds. To check if increased levels of BAMBI indeed inhibit the canonical (Smad2/3-dependent) TGF-β signaling pathway, the Ad5-CAGA-Luc reporter assay was repeated in the presence or absence of rhBAMBI. As control condition cells were stimulated with rhTGF-β1, which significantly induced Smad2/3 signaling. The addition of rhBAMBI dose-dependently reduced the active Smad2/3 signaling (Figure 7a). In line with this observation, the Smad2/3 signaling activated by the patients’ sera was significantly reduced by the addition of rhBAMBI (Figure 7b).

As indicated by the name, BAMBI was first identified as an inhibitor for BMP and activating (Smad1/5/8-dependent) signaling. Therefore, a BMP (Ad5-BRE-Luc) reporter assay was performed in the presence or absence of rhBAMBI. As expected, the addition of rhBMP7 significantly activated Smad1/5/8 signaling. The addition of rhBAMBI significantly inhibited Smad1/5/8 signaling—in contrast to the Smad2/3 signaling, the addition of rhBAMBI reduced Smad1/5/8 signaling even below the baseline without the addition of rhBMP7. The addition of rhTGF-β1 did not activate but inhibited Smad1/5/8 signaling (Figure 7c). In line with this observation, the Smad1/5/8 signaling was inhibited by the patients’ sera, especially the sera from patients with chronic wounds which contained high levels of TGF-β and BAMBI. The Smad1/5/8 signaling inhibited by the patients’ sera was further reduced by the addition of rhBAMBI (Figure 7d).

## 4. Discussion

This study aimed to investigate the angiogenic potential and associated TGF-β signaling in tissues of acute and chronic wounds more closely.

A major aspect of chronic or non-healing wounds is the reduced ability to regrow microvasculature through the process of angiogenesis. Therefore, pro- and anti-angiogenic factors were screened in both patients’ sera and wound tissues. Although, the patients’ sera suggest a shift from VEGF/VEGFR signaling to ANGPT/TIE2 signaling in chronic wounds, this shift could not be observed in the wound tissues or the in vitro scabs. Instead, the chronic wound tissues showed decreased levels of endostatin, which is one of the most potent known inhibitors of angiogenesis [21]. Thus, impaired angiogenesis cannot be assumed in these chronic wound tissues. This assumption is further underlined by an increased expression of MMP9 in the chronic wound tissues. MMP9 is a major chemokine regulator of wound healing [22], secreted by neutrophils and macrophages. MMP9 expression in macrophages is dependent on their polarization state—more precisely, it is associated with the M2 phenotype [23]. The elevated levels of TGF-β1 in the serum of patients with chronic wounds should favor M2 polarization of macrophages, however, the increased expression of pro-inflammatory interleukins indicates more M1 macrophages in the chronic wound tissues. In the in vitro scabs, MMP9 expression was shown to be induced by the addition of rhTGF-β, independent of the individual isoform [16]. This effect was not observed for its tissue inhibitors TIMP1 and TIMP2 [16], which tightly regulate MMP9 activity by binding to their active center in a 1:1 ratio [24]. This process blocks the binding of MMPs to ECM substrates. During wound healing, epithelial cells and fibroblasts express TIMPs, e.g., TIMP1 and TIMP2 [25]. From these two cell types, only epithelial cells (HaCaT cells) are represented in the in vitro scabs. In the chronic wound tissues, the expression of *MMP9* tended to be increased, whereas the expression of *TIMP1* and *TIMP2* was unaffected. The resulting shift in the MMP:TIMP ratio suggests pro-angiogenic effects.

However, the increased expression of MMPs might not only indicate active angiogenesis but also active degradation of the ECM. Substrates of MMP9, also known as gelatinase B, i.e., include different types of collagens, gelatinss, elastin, and TGF-β precursors [26]. Despite the increased expression of *MMP9*, expression of *CTGF* and *COL1A1* were not increased in chronic wound tissues. However, the expression of fibronectin was increased in the examined chronic wound tissues. Fibronectin has diverging effects on wound healing. During early wound healing, both plasma- and tissue-derived fibronectin provide a scab stabilizing matrix that supports re-epithelialization by interaction with α5β1 integrin [27,28]. During later stages of wound healing, this temporary fibronectin matrix will be remodeled and replaced by a collagenous matrix. Incomplete fibronectin clearance has been associated with impaired wound healing [29], which is in line with our results, where the highest fibronectin expression levels were detected in chronic wound tissues.

Fibronectin is well known to bind fibrin, whose fragment E has been identified as chemoattracting fibroblasts and potentiating TGF-β-induced myofibroblast activation [30]. An important role in this process might be the expression of MMP9, reported to be induced by fibronectin and vitronectin in different cell types, among others in HUVECs. Since increased levels of MMP9 were reported to favor the release and activation of TGF-β [31], this could explain both the increased expression of *MMP9* as well as the elevated levels of active TGF-β in the chronic wound tissues in our study.

Although TGF-β is required for successful wound healing, its regulation is critical. Interestingly, prolonged expression of *TGF-β1* by keratinocytes and immune cells was associated with both chronification of wounds [32] and excessive scar or keloid formation [33]. It has been speculated that the activation of TGF-β is crucial for the expected outcome. In our wound tissues, increased levels of active TGF-β were expected to induce the expression of ECM proteins, e.g., *CTGF* or *COL1A1*, while suppressing inflammation. However, this effect was absent, indicating that TGF-β signaling was blocked in the respective target cells. The assumption of reduced canonical TGF-β signaling was confirmed by Western blotting and immune staining showing reduced phospho-Smad3 levels both overall in the chronic wound tissues and in the nuclei of the resident cells. The overall reduction in phospho-Smad3 levels suggested that the TGF-β signaling is abrogated before the ligands can activate their receptors. This is further substantiated by the finding that *CTGF* expression, shown to be regulated by non-canonical TGF-β signaling, including Ras/MEK/ERK-MAPK [34], was not induced in the chronic wound tissues. CTGF then assists TGF-β to stimulate the expression of ECM proteins, e.g., collagen or fibronectin, in wound fibroblasts via the canonical TGF-β signaling cascade [35]. In the chronic wound tissues examined in our study, fibronectin, but not collagen expression was induced. The lack of subsequent collagen-guided fibronectin remodeling is associated with non-healing wounds [33]. In normal wound healing, recruited fibroblasts, keratinocytes, and endothelial cells get activated by TGF-β to undergo EMT, which in turn allows the cells to form collagen and produce angiogenic factors, e.g., VEGF [8].

In contrast, inhibition of the TGF-β signaling was assumed, justifying the search for TGF-β inhibitors. By using the RayBio^®^ Human TGF-β Array 2, elevated levels of the TGF-β pseudo-receptor BAMBI were detected in both sera and wound tissues of patients with chronic wounds. This presumption was confirmed by qRT-PCR, which further revealed increased *BAMBI* expression in acute wounds that chronified in the following healing process. So far, little has been reported in the literature on the role of BAMBI in the process of wound healing. According to the human protein atlas (https://www.proteinatlas.org/ENSG00000095739-BAMBI/single+cell+type/skin, accessed on 22 June 2023) BAMBI is expressed in human skin by melanocytes, granulocytes, and to a smaller extent by macrophages—with little to no expression in fibroblasts, keratinocytes, and endothelial cells. In macrophages, overexpression of BAMBI was reported to induce a shift from pro-healing M2 to pro-inflammatory M1 phenotype, including the proliferation of the latter [36], which could explain the increased expression of pro-inflammatory interleukins in chronic wound tissues. Although the histological staining showed inhomogeneous phospho-Smad3 staining, the overall reduced staining for phospho-Smad3 suggested that the inhibition of TGF-β signaling in the chronic wound tissues was not restricted to the immune cells. The immunohistological stainings illustrate one weakness of this study, which is the inhomogeneity of the investigated tissues. Further, canonical TGF-β involves not only phospho-Smad3 but also phospho-Smad2, which should be considered when addressing different target genes.

BAMBI is known as a pseudo-receptor for the TGFβRI family, scavenging free ligands of TGF-β from receptor binding, thus preventing the initiation of the following intracellular signaling. It has been discussed whether BAMBI needs to be membrane-bound to exert its inhibitory effects. In our experiments, the addition of recombinant human BAMBI dose-dependently inhibited the canonical TGF-β (Smad2/3) signaling. As indicated by the name, rhBAMBI also strongly inhibited the canonical BMP (Smad1/5/8) signaling. TGF-β itself, also interfered with the canonical BMP signaling—an effect previously observed in mesenchymal cells. In these cells, a TGF-β-dependent upregulation of Ski-related novel protein N (SnoN) was a crucial factor [37], which cannot be ruled out in the wound tissues. BAMBI itself was not identified as a Smad2 or Smad3 target gene within the Harmonize 3.0 database (https://maayanlab.cloud/Harmonizome/gene_set/SMAD2/CHEA+Transcription+Factor+Targets & https://maayanlab.cloud/Harmonizome/gene_set/SMAD3/CHEA+Transcription+Factor+Targets, both accessed on 10 August 2023).

However, a synergistic inhibitory effect of TGF-β and BAMBI on canonical BMP signaling can be proposed, as sera from patients with chronic wound, containing higher amounts of active TGF-β and BAMBI, more strongly inhibited Smad1/5/8 signaling than sera from patients with acute wound, containing less active TGF-β and BAMBI. BAMBI was reported to not only affect the canonical, Smad-dependent signaling but also the non-canonical signaling pathways [38], which might explain the lack of increase in both *CTGF* and *COL1A1* expression in the chronic wound tissues. In the present study, non-canonical TGF-β signaling was not addressed. As prolonged TGF-β signaling is normally associated with an excessive scar or keloid formation [33], modulation of BAMBI in experimental wound healing so far addressed the potential therapeutic effects of BAMBI. For example, BAMBI has been shown to inhibit skin fibrosis in keloids by suppressing TGF-β1-induced hyper normal fibroblast proliferation and excessive type I collagen accumulation [39]. Consistent with this finding, suppression of BAMBI by miR-519d-3p promoted TGFβ/Smad-mediated postoperative epidural scar formation [40]. Modulation of BAMBI, however, not only affected ECM formation. BAMBI knockout mice developed a phenotype characterized by endothelial activation and proliferation [41], resulting in accelerated re-endothelialization, increased neovascularization, and erythrocyte extravasation after arterial injury [42]. It was postulated that these effects require an alternative pathway activation by TGF-β through ALK1. In addition, TGF-β induced recruitment of hematopoietic effector cells, expression of *VEGF* [7] (confirmed by our in vitro data), or induction of EMT [8] might contribute to the improved re-vascularization in this model.

## 5. Conclusions

In summary, the herein presented data show reduced TGF-β signaling despite increased expression and activation of TGF-β in the examined chronic wound tissues when compared to acute wound tissues. As a possible regulatory mechanism, increased expression of the TGF-β pseudo-receptor BAMBI was identified.

## Figures and Tables

**Figure 1 cells-12-02095-f001:**
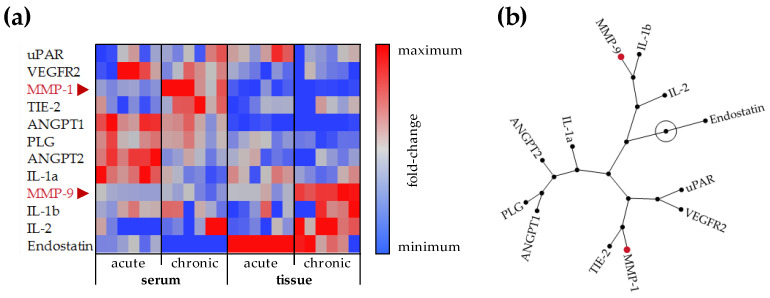
Angiogenic profile in wound tissues and sera of patients with acute and chronic wounds. The RayBio^®^ Human Angiogenesis Array C2 was performed with sera and lyzed wound tissues of patients with acute and chronic wounds. For each group, N = 12 patients were pooled for the analysis. The array was performed three times to obtain 6 individual data points (duplicates on each array). After quantification with ImageJ 1.47v, data points were analyzed with JMP 16. (**a**) Array data in the heat map represent fold changes for each individual factor. (**b**) The constellation plot shows correlations between the individual factors.

**Figure 2 cells-12-02095-f002:**
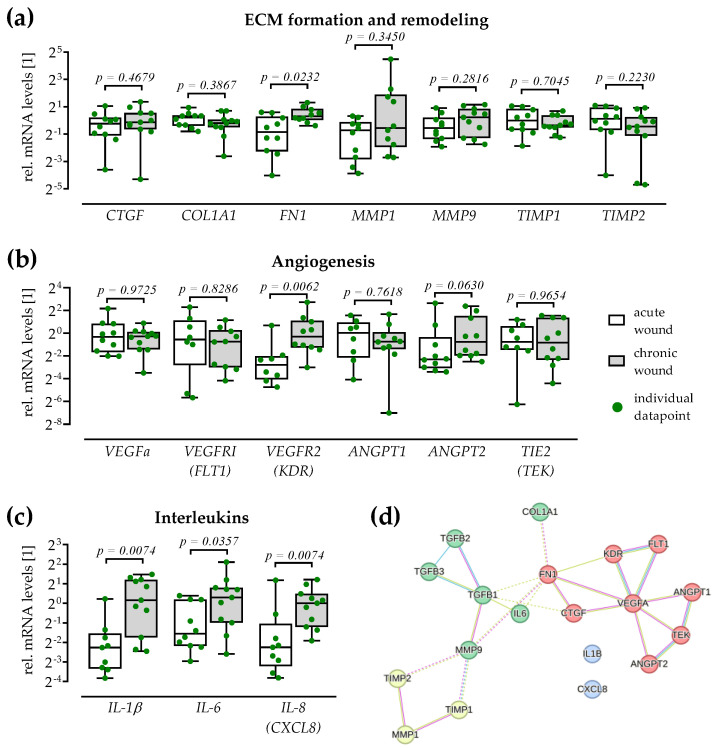
Gene expression in tissues from acute and chronic wounds with TGF-β as a common regulator. For each group (white: acute wounds/grey: chronic wounds), N ≥ 10 wound tissues were processed. RT-PCR was performed in duplicates (n = 2) with *18s* as housekeeper. Relative mRNA levels are presented as dot plots with individual data points. Groups were compared by non-parametric Mann–Whitney U tests. (**a**) Genes involved in matrix formation included *CTGF*: connective tissue growth factor, *COL1A1*: alpha-1 type I collagen, and *FN1*: fibronectin. Genes involved in matrix degradation included the matrix metalloproteinases *MMP1* and *MMP9*, as well as their tissue inhibitors *TIMP1* and *TIMP2*. (**b**) Genes relevant for angiogenesis included *VEGFa*: vascular endothelial growth factor A and its receptors *VEGFR1* (*FLT1*) and *VEGFR2* (*KDR*), as well as the angiopoietins *ANGPT1* and *ANGPT2* and their receptor *TIE2* (TEK tyrosine kinase or CD202B). (**c**) Further, interleukins IL-1β, IL-6, and IL-8 (CXCL8) were detected. (**d**) String analysis (https://string-db.org/cgi/network?taskId=bPNoneFVLRuc&sessionId=bRsopOdB43uA, accessed on 22 June 2023) identified TGF-β (transforming growth factor beta) as a common regulator in this network.

**Figure 3 cells-12-02095-f003:**
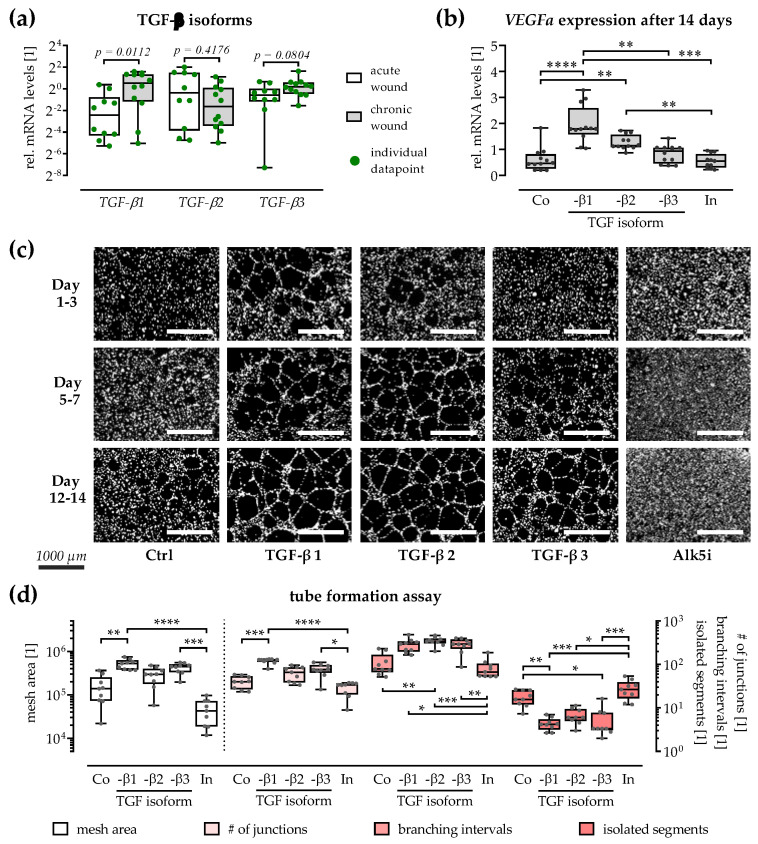
TGF-β isoforms are expressed in chronic wound tissue and induce *VEGFa* expression in vitro. (**a**) For each group (white: acute wounds/grey: chronic wounds), N ≥ 10 wound tissues were processed, and expression of all three TGF-β isoforms was quantified in duplicates (n = 2). Groups were compared by non-parametric Mann–Whitney U tests. (**b**) In vitro scabs (N = 6) were produced and stimulated with the three recombinant human TGF-β isoforms or the small molecule inhibitor for Alk5 (In.: SB431542) for up to 14 days. Expression of *VEGFa* (vascular endothelial growth factor A) was quantified in duplicates (n = 2). *18s* was used as housekeeper for the RT-PCRs. Culture supernatants from the in vitro scabs were collected after 3, 7, and 14 days and applied onto HUVECs for the tube formation assays (N = 3, n = 3). (**c**) Representative microscopic images from the Calcein-AM stained HUVECs from the tube formation assay. Scale bar = 1000 µm (**d**) For each group images from 5 pre-defined positions within the well were obtained and analyzed with the angiogenesis analyzer of ImageJ 1.47v, which provides information on the mesh area, the average number of junctions, the branching interval, and the number of isolated segments. Groups were compared by Kruskal–Wallis tests followed by Dunn’s tests for multiple comparisons. * *p* < 0.05, ** *p* < 0.01, *** *p* < 0.001, and **** *p* < 0.0001, as indicated. Data are displayed as dot plots with individual data points.

**Figure 4 cells-12-02095-f004:**
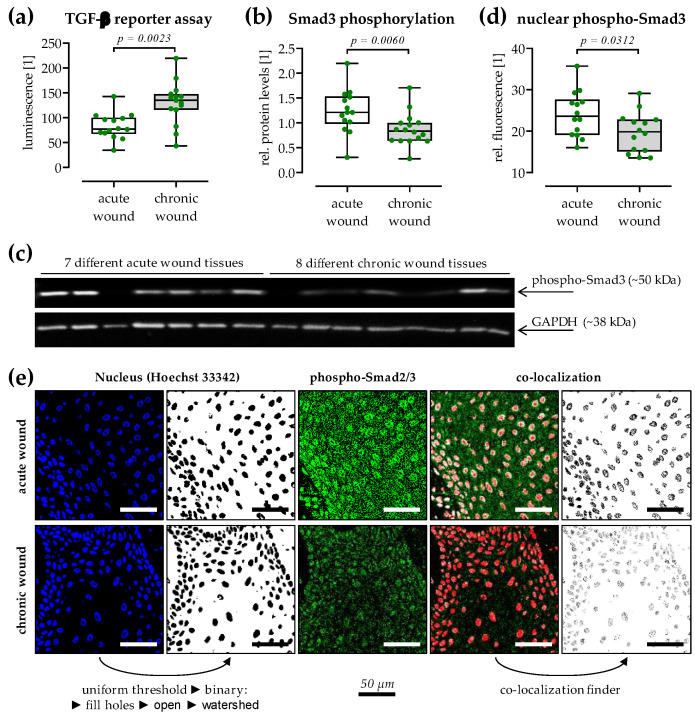
TGF-β in the serum of patients with chronic wounds is active but does not reach the wound tissue. (**a**) A TGF-β reporter assay (Ad5-CAGA-Luc in HaCaT cells) was performed using the collected sera from patients with acute (white) and chronic (grey) wounds. Tissues from the same patients were analyzed by (**b**,**c**) Western blot and (**d**) Immunofluorescent staining to quantify the overall phosphorylation of Smad3 and its nuclear levels. (**c**) Representative Western blot image. (**e**) Representative workflow for the analysis of the confocal images (n = 3 per slide) in ImageJ 1.47v. scale bar = 50 µm. Data are presented as box plots with individual data points. Groups were compared by non-parametric Mann–Whitney U test.

**Figure 5 cells-12-02095-f005:**
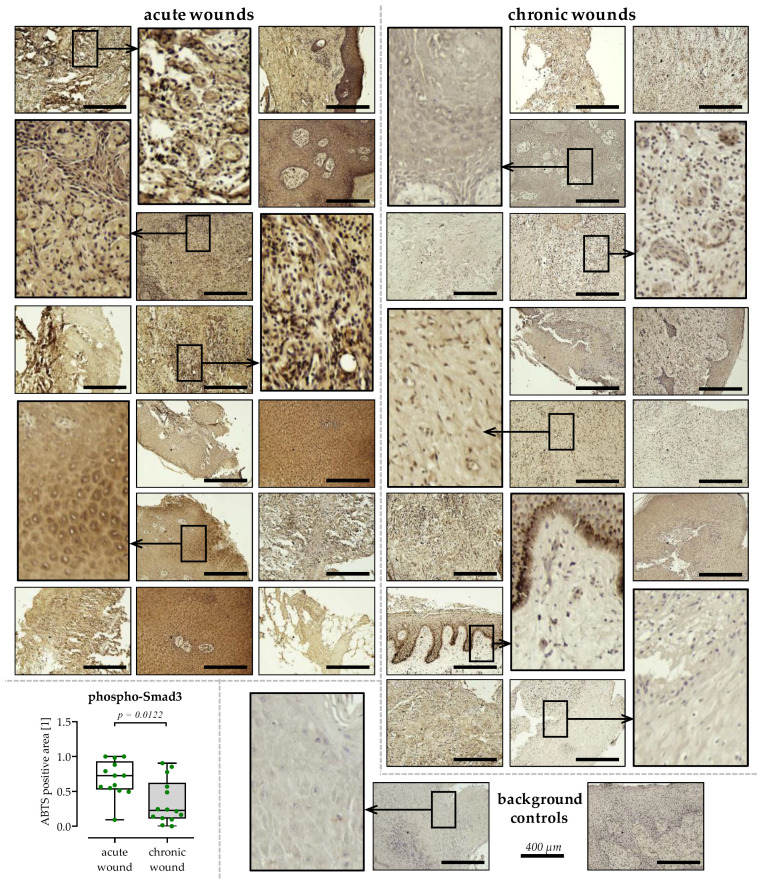
Phospho-Smad3 staining in tissues obtained from acute and chronic wounds. Immunohistochemical staining for phospho-Smad3 was performed (ABTS with Hematoxylin counterstain) to identify active TGF-β signaling within the wound tissues. N = 13 tissues from acute wounds (white) and N = 14 tissues from chronic wounds (grey) were stained, each with an individual background control (no primary antibody). Overview images are shown for each tissue captured with a 100-fold magnification, scale bar = 400 µm. For selected tissues, areas were also magnified. Image analysis with ImageJ 1.47v was performed using 3 images with a higher resolution (200-fold magnification) per tissue slide. Data, summarized as a dot plot with individual data points, were compared by non-parametric Mann–Whitney U test.

**Figure 6 cells-12-02095-f006:**
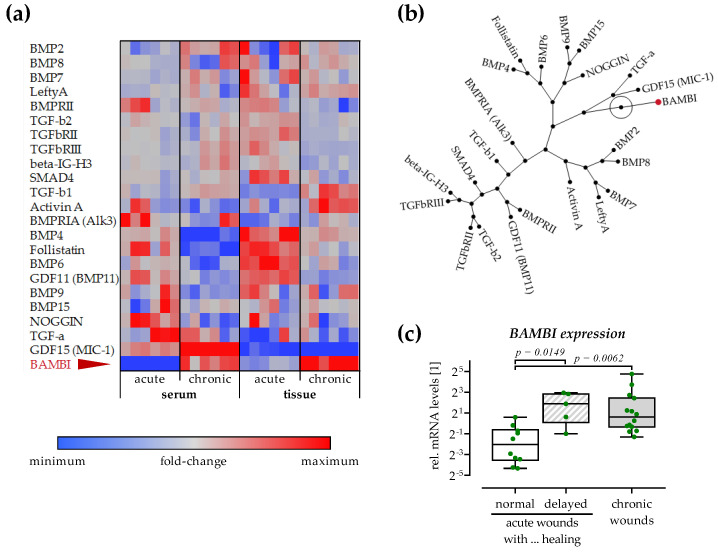
Increased BAMBI levels in wound tissues and sera of patients with acute and chronic wounds. The RayBio^®^ Human TGF-β Array 2 was performed with sera and lyzed wound tissues of patients with acute and chronic wounds. For each group, N = 12 patients were pooled for the arrays. The array was performed three times to obtain 6 individual data points (duplicates on each array). After quantification with ImageJ 1.47v data points were analyzed with JMP 16. (**a**) Array data in the heat map represent fold changes for each individual factor. (**b**) The constellation shows correlations between the individual factors. (**c**) qRT-PCR for *BAMBI* in wound tissue samples from patients with acute (N = 15/N = 10 normal and N = 5 delayed healing) and chronic (N = 14) wounds. Each sample was measured in duplicates (n = 2) with *18s* as housekeeper. Relative mRNA levels were determined by the ΔΔC_T_ method. Groups were compared by Kruskal–Wallis tests followed by Dunn’s tests for multiple comparisons.

**Figure 7 cells-12-02095-f007:**
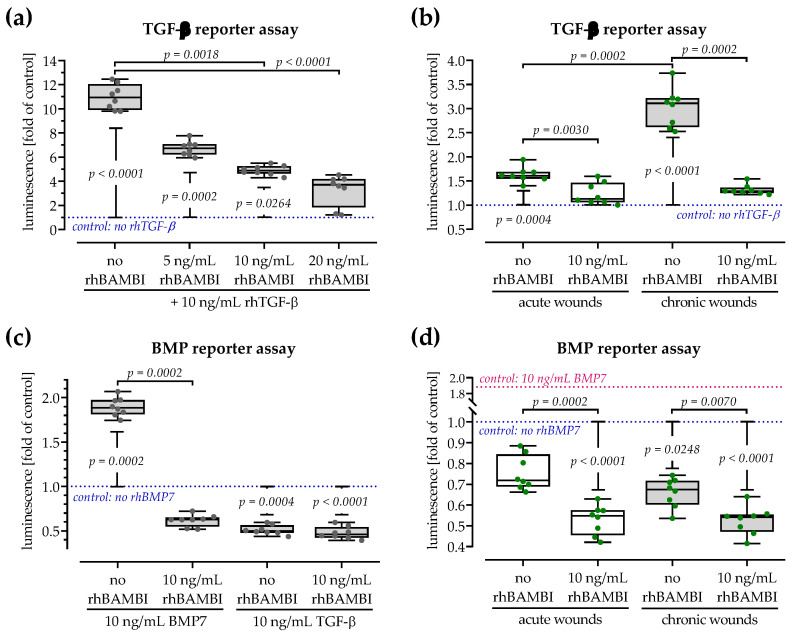
Recombinant human (rh) BAMBI inhibits both Smad2/3 and Smad1/5/8-dependent TGF-β signaling pathways. A TGF-β reporter assay (Ad5-CAGA-Luc in HaCaT cells) was performed using (**a**) 10 ng/mL rhTGF-β and increasing concentrations (5, 10, and 20 ng/mL) rhBAMBI, or (**b**) the collected sera from patients with acute (white) and chronic (grey) wounds in presence or absence of 10 ng/mL rhBAMBI. A BMP reporter assay (Ad5-BRE-Luc in HaCaT cells) was performed using (**c**) 10 ng/mL rhBMP7 or 10 ng/mL rhTGF-β, or (**d**) the collected sera from patients with acute (white) and chronic (grey) wounds—all in the presence or absence of 10 ng/mL rhBAMBI. For these experiments, the patients’ sera were pooled (each pool contained 12 serum samples). All measurements were performed as N = 4, n = 2. Data, summarized as dot plots with individual data points, were compared by non-parametric Kruskal–Wallis tests followed by Dunn’s tests for multiple comparisons and non-parametric Mann–Whitney U tests (1 condition ± BAMBI). Control conditions are presented as dotted lines.

**Table 1 cells-12-02095-t001:** Information on used primers and PCR conditions.

Primer	NM_	Forward Sequence	Reverse Sequence	T_A_	Size
*18s*	003286	GGACAGGATTGACAGATTGAT	AGTCTCGTTCGTTATCGGAAT	56 °C	111 bp
*ANGPT1*	001146.5	CGATGGCAACTGTCGTGAGA	CATGGTAGCCGTGTGGTTCT	60 °C	232 bp
*ANGPT2*	001147.3	CTTGGAACACTCCCTCTCGAC	GCTTGTCTTCCATAGCTAGCAC	60 °C	125 bp
*BAMBI*	012342.2	TGCACGATGTTCTCTCTCCT	GAAGTCAGCTCCTGCACCTT	59 °C	106 bp
*COL1A1*	000088.3	CAGCCGCTTCACCTACAGC	TTTTGTATTCAATCACTGTCTTGCC	60 °C	83 bp
*CTGF*	001901.3	CCAATGACAACGCCTCCTG	TGGTGCAGCCAGAAAGCTC	62 °C	159 bp
*FN1*	002026.2	CCCCATTCCAGGACACTTCTG	GCCCACGGTAACAACCTCTT	60 °C	203 bp
*IL-1β*	000576.2	CTCGCCAGTGAAATGATGGCT	GTCGGAGATTCGTAGCTGGAT	60 °C	144 bp
*IL-6*	000600.4	AACCTGAACCTTCCAAAGATGG	TCTGGCTTGTTCCTCACTACT	58 °C	159 bp
*IL-8 (CXCL8)*	000584.3	TAGCAAAATTGAGGCCAAGG	AAACCAAGGCACAGTGGAAC	60 °C	227 bp
*MMP1*	002421.4	CCCAGCGACTCTAGAAACACA	TCTTGGCAAATCTGGCGTGT	60 °C	322 bp
*MMP9*	004994.3	TCTATGGTCCTCGCCCTGAA	CATCGTCCACCGGACTCAAA	64 °C	219 bp
*RPL13a*	012423.3	AAGTACCAGGCAGTGACAG	CCTGTTTCCGTAGCCTCATG	56 °C	100 bp
*TGF-β1*	000660.6	TCCGGACCAGCCCTCGGGAG	CGGTCGCGGGTGCTGTTGTA	58 °C	680 bp
*TGF-β2*	001135599.1	GCAGGTATTGATGGCACCTC	AGGCAGCAATTATCCTGCAC	58 °C	206 bp
*TGF-β3*	003239.2	CAGCTGCCTTGCCACCCCTC	TGCAGCCTTCCTCCCTCTCCC	58 °C	601 bp
*Tie2 (TEK)*	000459.5	GGTCAAGCAACCCAGCCTTTTC	CAGGTCATTCCAGCAGAGCCAA	64 °C	121 bp
*TIMP1*	003254.3	AGTTTTGTGGCTCCCTGGAA	AAGCCCTTTTCAGAG-CCTTG	60 °C	179 bp
*TIMP2*	003255.5	ATGCAGATGTAGTGATCAGGGC	GTGATGTGCATCTTGCCGTC	60 °C	256 bp
*VEGFa*	001204384.1	CTACCTCCACCATGCCAAGT	GCAGTAGCTGCGCTGATAGA	59 °C	109 bp
*VEGFRI (FLT1)*	001160030.1	TCTCACACATCGACAAACCAATACA	GGTAGCAGTACAATTGAGGACAAGA	60 °C	106 bp
*VEGFRI (KDR)*	002253.2	CAGGGGACAGAGGGACTTG	GAGGCCATCGCTGCACTCA	60 °C	91 bp

T_A_ annealing temperature.

**Table 2 cells-12-02095-t002:** Information on the patients involved in this study.

	Acute Wounds		Chronic Wounds	*p*-Value *
	Healing	Delayed Healing		
# of patients	N = 15	N = 5	N = 18	-
males	N = 11	N = 2	N = 11	>0.9999
females	N = 4	N = 3	N = 7	
age (years)	52.9 ± 17.1 (18–82)	64.2 ± 11.8 (53–82)	58.3 ± 18.4 (20–81)	0.6182
wound duration at sampling (days)	22.4 ± 11.9 (0–37)	18.8 ± 9.2 (8–30)	203.8 ± 226.2 (64–730)	<0.0001
wound size at sampling (cm^2^)	33.3 ± 24.9 (8.3–96.0)	38.6 ± 29.8 (12.0–88.0)	32.7 ± 22.3 (2.3–80.0)	>0.9999
# of traumatic wounds	N = 6 (40.0%)	N = 3 (75.0%)	N = 8 (44.4%)	>0.9999
# of iatrogenic wounds	N = 2 (13.3%)	N = 0 (0.0%)	N = 3 (16.7%)	>0.9999
# of pressure/sheer wounds	N = 3 (20.0%)	N = 0 (0.0%)	N = 3 (16.7%)	>0.9999
# of wounds with other reasons°	N = 1 (6.7%)	N = 0 (0.0%)	N = 1 (5.6%)	>0.9999
# of wounds positive for germs	N = 7 (46.7%)	N = 3 (60.0%)	N = 12 (66.7%)	0.4323
# of infectious wounds	N = 3 (20.0%)	N = 1 (25.0%)	N = 3 (16.7%)	>0.9999
CRP (mg/L)	39.1 ± 70.3 (0.2–256.0)	4.1 ± 7.4 (0.2–17.3)	24.5 ± 37.7 (0.2–127.2)	0.4939
Neutrophil count (10^3^/µL)	4646 ± 1648 (1700–7130)	3654 ± 730 (2600–4360)	4104 ± 1516 (1990–7510)	0.9461
# of diabetics	N = 3	N = 2	N = 5	>0.9999
# of active smokers	N = 5	N = 2	N = 6	>0.9999
# of former smokers	N = 5	N = 2	N = 7	

* Mann–Whitney U test or Fisher’s exact test between acute and chronic wounds.

## Data Availability

The datasets generated during and/or analyzed during the current study are available from the corresponding author upon reasonable request.

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
