# Peer review of "Increased Levels of BAMBI Inhibit Canonical TGF-β Signaling in Chronic Wound Tissues"

_cells, 2023, doi:10.3390/cells12162095_

Round 1
Reviewer 1 Report
Major points:
1. Overall, the study is rather descriptive. I recommend to add at least some functional data on the role of BAMBI in the development of chronic wounds. For instance, adding either recombinant BAMBI protein or a neutralizing antibody to BAMBI to sera from patients with chronic wounds and testing these sera in their reporter assay for modulation of TGF-β signaling activity.
2. Line 365 (Legend to Fig. 5): Why didn’t the authors use an isotype IgG antibody as control instead of omitting the primary antibody?
3. The authors only studied phospho-Smad3. What about phospho-Smad2? Did the authors measure activation of this other R-Smad as well and were the results comparable with those of phospho-Smad3?
Minor points:
1. Please add the correct headline for Table 2.
2. The authors always compare acute and chronic wounds. What about a comparison between each of these wound types with normal skin?
3. In 3.4. The expression of TGF-β1 and TGF-β3 in chronic wound and acute tissues was compared by qPCR. Did the authors also measure and quantify the TGF-β isoforms at the protein level, i.e., by ELISA?
4. In the reporter gene assay the authors did not distinguish between the three TGF-β isoforms. Why not? Is this principally possible at all?
5. Line 333: Please rephrase this sentence as follows: …In line with the increased expression of active TGF-β1 and TGF-β3 in chronic wound tissues,…..because they did not measure total amounts of TGF-β1 and -3 but only the active portion of it.
6. Lines 336/337: The authors wrote: …. confirmed the result of the immunohistochemical staining (Fig. 4). However, the immunohistochemical data have not yet been presented at this stage but only appear in Fig. 5.
7. Line 349 (legend to Fig. 4): The authors wrote: “…and its translocation into the nucleus”. Please rephrase to: “…and its nuclear levels”, since they actually did not measure the translocation process itself but only its outcome (namely higher levels in the nucleus).
8. Line 476: a “p” is missing in presumtion (presumption).
9. Line 487: Remove the “r” in phosphor.
10. Line 503: The authors wrote: “It was postulated that these effects [in BAMBI knockout mice] require an alternative pathway activation by TGF-β through ALK1”. Since ALK1 signals through Smad1 and Smad5, it would be interesting to also study phosphorylation of Smad1/5 after modulation of BAMBI expression or activity.
I consider the manuscript by Ehnert et al to be of very high quality, technically sound, very innovative and of high interest to readers. As outlined in my comments, I only miss some functional data with respect to the BAMBI receptor. Based on my suggestions on the type of experiments the authors could perform, I am confident that these can be easily and rapidy performed by the authors.
The english is fine. Only minor spelling checks are required.
Author Response
We would like to thank the reviewer for the thorough review of our manuscript. The comments helped us to improve the quality of our manuscript. We performed additional experiments and addressed the points raised by the reviewer. Detailed answers to the reviewer's questions are attached.

Reviewer 2 Report
Although the study design features different methodologies (especially arrays) and the authors analyzed both serum samples and wound tissues from the same clinical cases, the presentation of results is not always easy to follow and the findings remain mainly descriptive. Such a descriptive nature of the study clearly reduces the enthusiasm of the reader.
Accordingly, the title "Increased levels of BAMBI impair TGF-β signaling in chronic wound tissues" is misleading, as the descriptive array findings and the validation of BAMBI expression only at the mRNA levels can only allow to hypothesize that this inhibitor impairs canonical TGF-β signaling in chronic wound tissues. Collectively, the true mechanistic implication of BAMBI in such a context remains to be fully demonstrated.
Another important point to consider is that TGF-β signaling includes both canonical and non-canonical pathways. The authors focused exclusively on the canonical, Smad3-mediated signaling. Since they found an inhibition of this pathway (reduced Smad3 phosphorylation and nuclear translocation) in the presence of activated TGF-β isoforms, they a priori searched for an inhibitor by employing a specific array. Unfortunately, the authors did not consider the non-canonical TGF-β signaling pathways (e.g. ERK1/2 phosphorylation and many others, such as Wnt/β-catenin, Notch1...), which could be activated in the chronic wound tissues.
Other important limitations of the study include:
The different array data (both angiogenesis and TGF-β pathway) were validated only by qRT-PCR analysis to measure mRNA expression levels of relevant targets. This is not sufficiently informative. Relevant targets should be confirmed at the protein levels by using Western blot on wound tissues and ELISA on serum samples.
The authors performed Western blot to assess phosphorylated Smad3. However there are no representative phosphorylated Smad3 bands of such an analysis in Figure 4, nor it is shown the loading control as well as total Smad3.
Phosphorylated-Smad3 immunostaining shown in Figure 5 appears mostly unspecific with very strong background. Moreover, very low magnifications are shown. Collectively, these immunohistochemical data are unreliable and requires extensive revisions.
Much work using in vitro and in vivo models is required to extend the present descriptive data and demonstrate that BAMBI is an important player in chronic wounds.
The Discussion section is poorly attractive and, because of the descriptive nature of the study, it contains too many speculations. The authors have discussed many literature findings which are not directly related to the findings of the present study.
Minor editing of English language and corrections of a few typos is recommended.
Author Response

(The authors gave the same response as above.)

Reviewer 3 Report
In my opinion, the work entitled “Increased levels of BAMBI impair TGF-β signaling in chronic wound tissues” is very interesting. It deals with a topic of great clinical interest such as wound repair.
The introduction and discussion are very clear, they masterfully explain the implication of TGF-beta in healing processes, and clarify the results of the studies, in line with the literature data.
The results are clear and well structured.
I have only one observation: in the introduction the importance of BAMBI is not highlighted and since the work is focused on the increased levels of this inhibitor, I suggest to add a little more detailed description of BAMBI in this section.
Author Response
We would like to thank the reviewer for the thorough review of our manuscript. The comments helped us to improve the quality of our manuscript. We performed additional experiments and addressed the points raised by the reviewer. The answer to the reviewer's question is attached.

Reviewer 4 Report
Manuscript tackles an intriguing aspect of TGF-beta signaling's involvement in the pathophysiology of chronic and acute wound healing. It offers valuable insights into the molecular mechanisms related to chronic wounds and highlights the potential regulatory role of the TGF-β pseudo-receptor BAMBI in their development. The study presents numerous original results, reinforcing its significance for publication. However, I have a few minor comments. Firstly, it would be beneficial to include a scale in the tissue micrographs for improved visualization. Secondly, expanding the information on BAMBI in the introduction would enhance the reader's comprehension of its importance. Thirdly, discussing the reliability and limitations of the "scabs model" in the discussion section would fortify the manuscript. Additionally, relating the results of gene expression to healthy skin tissue would provide a valuable comparative analysis. Lastly, the conclusions should be derived directly from the presented research, rather than relying on the findings of other authors. Therefore, I suggest incorporating this part into the discussion section.
Author Response

(The authors gave the same response as above.)

Round 2
Reviewer 1 Report
The authors have satisfactorily responded to all my concerns. The ms. is now suitable for publication.
Dear Editor, dear Elsa,
the authors have satisfactorily responded to all my concerns. The ms. is now suitable for publication.
Author Response
We are happy the reviewer liked the modified version of our manuscript.
Reviewer 2 Report
The authors have revised their manuscript taking into account my comments.
There are only a few points that still require revision:
- There are errors in the identifications of the different panels of figure 4 in the legend. For instance, panel “c” shows representative western blot bands, not immunofluorescence images as reported in the figure legend.
- The following issue requires particular attention: the authors used GAPDH as loading control to normalize protein expression levels of phospho (p)-Smad3. This is not correct. The authors should measure the levels of total Smad3. This is very important to provide reliable data and clear demonstration of canonical, Smad3-dependent TGF-beta signaling activation. Of concern, there is strong variability in GAPDH expression among different samples, and variation in p-Smad3 levels seems to perfectly reflect that of GAPDH (i.e. differences in sample loading).
Author Response
We would like to thank you for the opportunity to revise our manuscript “Increased levels of BAMBI impair TGF-β signaling in chronic wound tissues” a second time for Cells.
We would like to thank reviewer 2, who noticed the mistake in the legend to figure 4. We have changed the figure legend to fit the modified figure 4. Answeres to the raised points are attached.
As the reviewers have suggested that there are no or only minor changes in the English required, we have used the Grammarly software (Grammaly, Inc.; San Francisco, CA, USA) to double check the English.
